

# A novel ddPCR™ assay for eDNA detection and quantification of Greater Amberjack *Seriola dumerilli* and three congeners in US waters: challenges and application to fisheries independent surveys

P. Joana Dias[1], Ryan Lehman[1], Bryan L. Huerta-Beltrán[1],
Ana Wheeler[2,3], Crystal L. Hightower[4,5], Jessica Heise[6],
Theodore Switzer[6], Clint Rhode[7], J. Marcus Drymon[2,3], Lynne Stokes[8],
Mark A. Albins[4,5], Sean Powers[4,5] and Nicole Phillips[1]

[1] School of Biological, Environmental, and Earth Sciences, University of Southern Mississippi, Hattiesburg, MS, United States
[2] Mississippi-Alabama Sea Grant Consortium, Ocean Springs, MS, United States
[3] Coastal Research and Extension Center, Mississippi State University, Biloxi, MS, United States
[4] Stokes School of Marine and Environmental Sciences, University of South Alabama, Mobile, AL, United States
[5] Dauphin Island SeaLab, Dauphin Island, AL, United States
[6] Fish and Wildlife Research Institute, Florida Fish and Wildlife Conservation Commission, St Petersburg, FL, United States
[7] Department of Genetics, Stellenbosch University, Stellenbosch, South Africa
[8] Department of Statistical Science, Southern Methodist University, Dallas, TX, United States

Corresponding author
P. Joana Dias,
joanadias77@hotmail.com

## ABSTRACT

**Background:** Four *Seriola* species support recreational and commercial fisheries along the U.S. Atlantic Ocean and the Gulf of Mexico, with the *S. dumerili* Gulf of Mexico stock being overfished for over three decades. The study presented here is part of a fisheries-independent project initiated to determine an absolute abundance of *S. dumerili*, to expand biological knowledge of the species and to develop novel tools for fisheries management. Environmental DNA (eDNA) tools aimed at the detection and quantification of target species are starting to emerge in support of marine fisheries surveys. Key to progressing the field is Droplet Digital™ PCR (ddPCR™), a highly sensitive technique with advanced multiplexing and direct quantification capabilities that can provide fisheries scientists with improved interpretation of eDNA data.

**Methods:** We developed and validated a novel tetraplex ddPCR™ assay able to detect and distinguish between *S. dumerili*, *S. fasciata*, *S. rivoliana*, and *S. zonata* from seawater eDNA samples. In order to groundtruth ddPCR™ data, and explore its capacity to provide abundance estimates, we compared ddPCR™ detections and quantifications to abundance data inferred from multiple camera (ROV, S-BRUV, chevron trap) and acoustic (VPS array) gears deployed during a fisheries research gear-calibration cruise.

**Results:** We demonstrated that with eDNA contamination controls and best practice protocols, it is viable to conduct eDNA research as part of a fisheries survey cruise. eDNA sampling was completed in less time than camera gears (15 min *vs* 2 h). Both
eDNA and camera gears detected the presence of *S. dumerili* and *S. rivoliana* at both sites and all sampling days, but not *S. fasciata* and *S. zonata*. eDNA concentration data was higher for *S. dumerili* than *S. rivoliana* at both sites for all sampling days, in line with abundance patterns obtained from camera gears. The highest correlation (r = 0.97) was obtained between the measures of eDNA between gear deployments and ROV.

**Discussion:** Incorporating eDNA in fisheries surveys would not require additional days at sea and could improve precision in fish detection and abundance. eDNA can be a valuable complement to camera gears deployed in geographic areas or seasons with poor visibility conditions, where fish may be present but cannot be confidently identified to the species level. The high correlation obtained between ROV and eDNA data collected between gear deployments adds to a growing number of studies demonstrating the potential of eDNA as an indicator of abundance for fisheries stock assessments. Time-series data from a carefully designed eDNA survey, that estimates relative abundance, could be used as an index of relative abundance for the *S. dumerili* stock assessment. To achieve this, investment into follow-up studies with increased sample sizes and spatial and temporal replication would be necessary to allow for year-to-year comparisons and validate the robustness of the correlation observed.

# INTRODUCTION

The genus *Seriola* (Perciformes, Carangidae) comprises nine recognized species of ecologically and economically important pelagic epi-benthic reef fishes with circumglobal, subtropical-temperate distributions (*Swart et al., 2015*). In U.S. waters, four species—greater amberjack *S. dumerili*, lesser amberjack *S. fasciata*, almaco jack *S. rivoliana*, and banded rudderfish *S. zonata*—occur along the U.S. Atlantic Ocean and the Gulf of Mexico, where they have supported recreational and commercial fisheries since the 1950s (*Berry & Burch, 1979*; *Galbraith et al., 2022*). *Seriola dumerili* promotion as a gamefish resulted in an increase in landings in the U.S. from three metric tons in 1962 to 607 metric tons in 1994 (*Cummings & McClellan, 1999*). Despite concerns regarding underestimates of catches (*Berry & Burch, 1979*), regional landings data in the U.S. were unavailable until 1991, with declines observed in commercial and recreational landings thereafter (*Harris et al., 2007*; *Southeast Data, Assessment and Review (SEDAR), 2020*). *Seriola dumerili* in the U.S. is currently managed as two discrete stocks under the Gulf of Mexico Fishery Management Council and the South Atlantic Fishery Management Council. Despite rebuilding efforts, the Gulf of Mexico stock is classified as overfished and undergoing overfishing for over three decades (*Southeast Data, Assessment and Review (SEDAR), 2020*).

Historic misidentification of the four species has further been an impediment to effective fisheries management (*Berry & Burch, 1979*; *Renshaw & Gold, 2009*; *Galbraith*

*et al., 2022*). Although taxonomic status of *S. dumerili*, *S. fasciata*, *S. rivoliana*, and *S. zonata* has been reasonably established based on the morphological identification of adults, similarities in their shape, coloration, and ontogenetic changes make them difficult to distinguish at earlier life stages. Recent analysis of Northeast Fisheries Science Center bottom trawl surveys conducted between 1963 and 2019 estimated at-sea assignments to be ≥91% correct for *S. fasciata*, *S. rivoliana*, and *S. zonata*, but only 24% correct for *S. dumerili*, making historic records of *S. dumerili* catches unreliable (*Galbraith et al., 2022*). Management decisions for *S. dumerili* have been contentious and therefore, a fisheries-independent study was initiated to determine an absolute abundance of *S. dumerili* in the U.S. Atlantic Ocean and Gulf of Mexico, to expand our biological knowledge of the species, and to develop novel tools to better inform fisheries management.

After years of attracting attention as a promising tool (*Hansen et al., 2017*), environmental DNA (eDNA) is beginning to be explored with the experimental dimension necessary to improve the cost-efficiency of fisheries research survey work (*Kelly et al., 2023*; *Saborido-Rey et al., 2023*). eDNA provides a non-invasive alternative for surveying aquatic communities (metabarcoding) with a demonstrated capacity to inform species biodiversity and distribution, which is of fundamental importance in ecosystem-based approaches (*Kelly et al., 2023*; *Saborido-Rey et al., 2023*). It can also provide species-specific (probe-based) detections with the potential to estimate abundance and biomass (*Rourke et al., 2022*), which are relevant parameters to stock management. However, to date, very few species-specific assays have been developed and applied to the detection and quantification of target species' eDNA from studies that parallel marine fisheries surveys (*Kirtane et al., 2021*; *Shelton et al., 2022*; *Ledger et al., 2024*; *Maes et al., 2024*).

The reliable capture and detection of eDNA in highly dynamic coastal and open-ocean environments and the use of eDNA to estimate fish abundance and biomass remain two of the most challenging goals of marine eDNA research (*Yates, Fraser & Derry, 2019*; *Ramírez-Amaro et al., 2022*; *Rourke et al., 2022*; *Kelly et al., 2023*). eDNA shed rate, transport and degradation are complex and are influenced by several biotic factors such as spawning season and animal size, and abiotic factors such as currents, water temperature, salinity, oxygen and pH (*Jerde, 2019*; *Takeuchi et al., 2019*; *Lamb et al., 2022*). Because eDNA shed rate is taxa and life-stage specific, and eDNA transport and degradation vary depending on season and geographic area, understanding these processes in each particular fisheries context is essential to plan efficient sampling, circumvent challenges and meet research aims. The need for more empirical studies (with increased temporal sampling and replication) on how eDNA in marine environments correlates to the presence, abundance, and biomass of fish stocks has, therefore, been pointed as the biggest caveat to progressing the field of eDNA research in applied fisheries science (*Kirtane et al., 2021*; *Maes et al., 2024*).

Key to progressing integration of eDNA as a fisheries management tool is also Droplet Digital™ PCR (ddPCR™), which has emerged as an improved, user-friendly, and cost-efficient alternative to real-time PCR (qPCR), with increased multiplexing capabilities, and demonstrated advantages in sensitivity and quantification (*Nathan et al.,*

*2014*; *Doi et al., 2015*; *Wood et al., 2019*; *Huggett & The dMIQE Group, 2020*). DdPCR™ partitions samples into thousands of nanoliter droplets (~20,000), with PCR-amplification of target DNA occurring within each individual droplet. DdPCR™ software quantifies the number of positive and negative droplets for each fluorophore in each sample and fits the fraction of positive droplets to a Poisson algorithm to determine the starting concentration of the target DNA molecule in units of copies/μL (*Huggett & The dMIQE Group, 2020*). By providing direct quantification, ddPCR™ has the potential to provide fisheries scientists with improved and straightforward interpretation of eDNA data. Despite the advantages, only one study has been published to date applying ddPCR™ to marine fisheries research (*Maes et al., 2024*) and none exploring its advanced multiplexing capabilities.

In the present study, we aimed to (1) develop and validate a tetraplex ddPCR™ assay able to detect and distinguish between *S. dumerili*, *S. fasciata*, *S. rivoliana*, and *S. zonata* in seawater eDNA samples. By comparing ddPCR™ detections and quantifications to relative abundance data inferred from multiple camera and acoustic gears deployed during a fisheries research gear-calibration cruise, we further aimed to (2) groundtruth ddPCR™ data, and (3) explore its capacity to provide abundance estimates and support fisheries independent surveys. By detailing assay development and validation, best practices in the prevention of contamination, and results of basic *in situ* experiments in the field, we also hope to provide useful guidance for future studies using ddPCR™ applied to marine fisheries eDNA research.

## MATERIALS AND METHODS

### eDNA contamination controls and best practice protocols

All work was conducted under strict procedures to reduce the risk of contamination by exogenous DNA and cross-contamination between samples in both the field and laboratory (*Goldberg et al., 2016*; *Bockrath et al., 2022*). All genomic DNA (gDNA) and eDNA laboratory work was temporally and spatially separated, and followed a unidirectional workflow. gDNA and eDNA samples were extracted in separate laboratories and all gDNA, synthetic DNA (sDNA), and eDNA samples, extracts, and aliquots were stored in separate rooms. All PCR reagents and aliquots were stored in a dedicated freezer, separate from all DNAs. Reaction setup for gDNA and sDNA PCR amplifications were performed under a PCR hood in a separate room from the PCR hood and ddPCR™ equipment used to perform eDNA PCR amplifications. Separate pipette sets were maintained in each laboratory space, with project-specific pipette sets dedicated to eDNA extractions and PCR setups. Barrier filter tips were used when conducting all reagent aliquoting, PCR, and eDNA work. Personal hygiene (showering and changing clothes between laboratory days and between working with gDNA, sDNA or eDNA), changing of disposable gloves, and deep cleaning laboratory spaces between temporally separate project phases (gDNA, sDNA, eDNA) were also implemented as a best practice. Benches were regularly cleaned with 10% bleach (before, between, and after work), all reusable equipment was soaked in 10% bleach for at least 20 min and rinsed thoroughly (*e.g.*, 3×) with deionized water, and all sensitive equipment was cleaned with DNA AWAY™

Surface Decontaminant (Thermo Scientific, Waltham, MA, USA), or a CloroxPro™ Healthcare® Bleach Germicidal Wipes and UV sterilized for at least 20 min.

All field equipment was handled, cleaned, and stored in a dedicated room in a separate building from the genetic work. All water collection and filtering equipment were cleaned ahead of fieldwork and between uses in the field using a combination of two methods of sterilization: cleaning with 10% bleach followed by either autoclaving at 120 °C for 20 min or exposure to UV light for 20 min, depending on the materials. Personal hygiene between fieldwork days (showering, clean clothes), changing of disposable gloves and separation of contamination-prone and clean areas both at sea and on shore were maintained throughout field activities (*Drymon et al., 2021*; *Lehman et al., 2022*). No fieldwork equipment was exposed to any individuals of the target species for the duration of this study. Negative controls were incorporated into each stage of sample processing as detailed in the following sections.

## Development of a tetraplex ddPCR™ assay
### Design and specificity

One set of degenerate primers (Eurofins Scientific, Luxemburg, Europe) and four species-specific internal PrimeTime® double-quenched ZEN™/IOWA Black™ FQ probes (Integrated DNA Technologies, Coralville, IA, USA) were manually designed in two mitochondrial DNA (mtDNA) loci—cytochrome oxidase c subunit 1 (COI), and NADH dehydrogenase 5 and 6 (ND5-ND6)—of the four target species (*Klymus et al., 2020*; BioRad Droplet Digital PCR Applications Guide Bulletin 6407). Briefly, sequences of the four target species and genetically similar co-occurring fish species were downloaded from GenBank and aligned with BioEdit© (*Hall, 1999*; *Alzohairy, 2011*). Primers were designed in regions conserved for the four target species but variable across other genetically similar and co-occurring fish species, while probes were designed to include base pair (bp) differences between all species. All primer and probe sets were checked for oligo dimers using the Thermo Fisher Multiple Primer Analyzer web tool and checked for specificity *in silico* using NCBI BLAST (*Altschul et al., 1990*; *Syngai et al., 2013*).

All primer sets were first tested for specificity *in vitro* with qPCR on a Bio-Rad® C1000 Touch™ Thermal Cycler (CFX96 Optics Module, instrument no. 785BR21404) using 10-fold dilutions starting from 5–25 ng of DNA (undiluted, 1:10, 1:100, 1:1,000) of gDNA extracted from one individual of each of the four target species and from one of each of the 25 exclusion species (Table 1). Reactions contained 1X Power SYBR® Green PCR Master Mix (Life Technologies LTD, Renfrew, UK) and 200 nanomolar (nM) of each primer, adjusted to 20 µL using PCR-grade water. Cycling conditions consisted of 95 °C for 10 min and 44 cycles of 94 °C for 15 s, 55 to 60 °C for 30 s and 60 °C for 1 min, followed by melting curve analysis between 65 °C and 95 °C with a plate read every 0.5 °C after holding the temperature for 5 s. The primer set that better amplified DNA from the target species, but not from exclusion species, was subsequently tested on all target and exclusion individuals with the respective species-specific probes (Table 2) using the optimized ddPCR™ conditions (details below). All qPCR and ddPCR™ reactions, including a non-template control (NTC, PCR-grade water added instead of template) were conducted in duplicate.

**Table 1 Name and origin of samples of target (grey shading) and non-target (exclusion) species used during assay development.**

| Type | Family | Name | Origin | Source |
|---|---|---|---|---|
| Target | Carangidae | Greater amberjack *S. dumerili* (8) | Alabama, Gulf of Mexico | This study |
| | | Greater amberjack *S. dumerili* (9) | | |
| | | Greater amberjack *S. dumerili* (11) | | |
| | | Greater amberjack *S. dumerili* (12) | | |
| | | Greater amberjack *S. dumerili* (13) | | |
| | | Almaco jack *S. rivoliana* (60) | Mississippi, Gulf of Mexico | |
| | | Almaco jack *S. rivoliana* (61) | | |
| | | Almaco jack *S. rivoliana* (62) | | |
| | | Lesser amberjack *S. fasciata* (1) | Louisiana, Gulf of Mexico | |
| | | Lesser amberjack *S. fasciata* (2) | | |
| | | Banded rudderfish *S. zonata* (3) | North Carolina, USA | |
| | | Banded rudderfish *S. zonata* (4) | | |
| | | Greater amberjack *S. dumerili* (Sdu9) | Cape Lambert, Australia | *Swart et al. (2015)* |
| | | Greater amberjack *S. dumerili* (Sdu15) | Madeira, Northwest Africa | |
| | | Greater amberjack *S. dumerili* (Sdu611) | Key Largo, FL, USA | |
| | | Lesser amberjack *S. fasciata* (Sfa3) | John's Island, SC, USA | |
| | | Lesser amberjack *S. fasciata* (Sfa10) | Cape Hatteras, NC, USA | |
| | | Almaco jack *S. rivoliana* (Sri3) | John's Island, SC, USA | |
| | | Almaco jack *S. rivoliana* (Sri36) | South Carolina, USA | |
| | | Almaco jack *S. rivoliana* (Sri37) | South Carolina, USA | |
| | | Banded rudderfish *S. zonata* (Szo2) | Panama City, FL, USA | |
| | | Banded rudderfish *S. zonata* (Szo20) | South Carolina, USA | |
| Exclusion | | Guinean amberjack *S. carpenteri* (Scar2) | Angola, West Africa | |
| | | Samson fish *S. hippos* (Ship1) | Eyre Peninsula, Australia | |
| | | Samson fish *S. hippos* (Ship2) | New South Wales, Australia | |
| | | Samson fish *S. hippos* (Ship5) | Rottnest, Australia | |
| | | Yellowtail amberjack *S. lalandi* (A1) | Southwest rift, South Africa | |
| | | Yellowtail amberjack *S. lalandi* (Slala2) | New South Wales, Australia | |
| | | Yellowtail amberjack *S. lalandi* (Slala3) | New South Wales, Australia | |
| | | Japanese amberjack *S. quinqueradiata* (Squi1) | Kochi Prefecture, Japan | |
| | | Japanese amberjack *S. quinqueradiata* (Squi2) | Kochi Prefecture, Japan | |
| | | Japanese amberjack *S. quinqueradiata* (Squi21) | Kochi Prefecture, Japan | |
| | | Crevalle jack *Caranx hippos* | Gulf of Mexico | This study |
| | | Horse-eye jack *Caranx latus* | | |
| | | Florida pompano *Trachinatus carolinus* | | |
| | | Permit *Trachinatus falcatus* | | |
| | | Rough scad *Trachurus lathami* | | |
| | | Round scad *Decapterus punctatus* | | |
| | | Rainbow runner *Elagatis bipinnulata* | | |
| | | Atlantic moonfish *Selene setapinnis* | | |

| Type | Family | Name | Origin | Source |
|------|--------|------|--------|--------|
| | Other | Greater barracuda *Sphyraena barracuda* | | |
| | | Cobia *Rachycentron canadum* | | |
| | | King mackerel *Scomberomorus cavalla* | | |
| | | Swordfish *Xiphias gladius* | | |
| | | Gag grouper *Mycteroperca microlepis* | | |
| | | Red snapper *Lutjanus campechanus* | | |
| | | Tilefish *Lopholatilus chamaeleonticeps* | | |
| | | Tripletail *Lobotes surinamensis* | | |
| | | Red drum *Sciaenops ocellatus* | | |
| | | Gulf menhaden *Brevoortia patronus* | Bait | |
| | | Atlantic mackerel *Scomber scombrus* | | |
| | | Squid | | |
| | | Canned tuna sample | Sandwich on board | |

**Note:**
Tissue samples of local co-ocurring non-target species were sourced from co-authors research cruises. Bait and tuna sandwich tissue samples were collected during the gear calibration cruise onboard the *R/V E. O. Wilson*. DNA sub-samples of target and non-target *Seriola spp* obtained from *Swart et al. (2015)* are listed with the original coding (in brackets) as per table in their published work.

**Table 2 Gene region, name and sequence of oligos (primers and probes) of the developed assay with corresponding nucleotides size (nt), GC content and melting temperature (Tm).**

| Gene | Name | Oligos | Size (nt) | GC (%) | Tm (°C) | Amplicon size and Fluorophores |
|------|------|--------|-----------|--------|---------|-------------------------------|
| ND5_ND6 (mtDNA) | JackFw | CAACRTYCAACGRGGTATR | 19 | 47.4 | 60.3 | 124 bp |
| | JackRev | CGTGGGTTCTTCTYTTGAC | 19 | 50 | 60.3 | |
| | PSri | TCTCTCTTCCTGCTCACCCTTG | 22 | 54.5 | 67.3 | IDT SUN |
| | PSzo | ATTCTATTAGTCACCCACTAGACAGCTC | 28 | 42.9 | 64 | IDT SUN |
| | PSdu | CAAAACCTACCTCTCTCTCTTCCTACTTACC | 31 | 45.2 | 67 | FAM |
| | PSfa | CCTCGTCCTCATGGTCTTATTAGTTACC | 28 | 46.4 | 67.4 | FAM |

**Note:**
Primers amplified 124 bp of the gene region and probes were tagged with FAM and IDT SUN (VIC equivalent) fluorophores.

All qPCR and ddPCR™ runs were performed separately on target and non-target species, with no positive controls added to prevent cross-contamination.

### ddPCR™ optimization and NTC threshold

DdPCR™ reactions and cycling conditions were optimized using gDNA for each of the four target species and amplitude multiplexing on the Bio-Rad® QX200™ AutoDG™ Droplet Digital™ PCR System (Automated Droplet Generator instrument no. 773BR3222; T100™ Thermal Cycler instrument no. 621BR18911; QX200™ Droplet Reader instrument no. 771BR1496; Bio-Rad, Hercules, CA, USA). The QX200 droplet reader enables detection of fluorescence in two different channels (FAM and VIC/HEX). Primer and probe concentrations, annealing temperature, ramp rate, and cycle number were adjusted to produce positive clusters of droplets with high relative fluorescence units (RFUs), optimal separation of clusters corresponding to each pair of probes within each

channel, and little to no droplet "rain" (*Huggett & The dMIQE Group, 2020*; *Lehman et al., 2020*; *Kokkoris et al., 2021*). While other multiplexing strategies are available, tetraplexing using fluorescence amplitude makes pipetting easier and provides a clearer output (*von Ammon et al., 2022*). The optimized ddPCR™ reactions contained 1X Bio-Rad® ddPCR™ Multiplex Supermix for probes, 910 nM of each primer, 227 nM of each probe, and 1 µL of ~0.2 ng DNA from each of the four species, adjusted with PCR-grade water to a final volume of 22 µL. Using an automated droplet generator, 20 µL of each ddPCR™ reaction was combined with ~70 µL of automated droplet generation oil for probes to create up to 20,000 nanoliter-sized droplets prior to PCR, as per the manufacturer protocol (BioRad Droplet Digital™ PCR Applications Guide Bulletin 6407). Optimal ddPCR™ cycling conditions consisted of an initial step at 95 °C for 10 min, followed by 40 cycles of: 94 °C for 30 s and 61 °C for 2 min and a final step at 98 °C for 10 min, using a ramp rate of 1 °C/s.

To avoid incorrectly calling artifacts (*i.e.*, droplets in the absence of target DNA resulting from assay components interacting in a way that causes premature probe cleavage), a plate with 48 NTC ddPCR™ reactions was run. This ddPCR™ run was prepared using freshly made aliquots of all reagents, after all gDNA samples were archived at −80 °C and all laboratory spaces and equipment were deep cleaned. A NTC threshold and an optimal range of the positive droplet population were established for the optimized assay (*Lehman et al., 2020*; *Kokkoris et al., 2021*, Fig. 1). All ddPCR™ data were analyzed using Absolute Quantification (ABS), the QX Manager 1.2 Standard Edition software and the guidelines for amplitude multiplexing on the BioRad Droplet Digital™ PCR Applications Guide Bulletin 6407.

### DdPCR™ limit of quantification and limit of detection

Assay sensitivity is determined by its limit of quantification (LoQ), indicated by the lowest target copy number in a sample that can be reliably quantified with an acceptable level of precision and accuracy, and limit of detection (LoD), which defines the assay's ability to detect the target sequence at low levels (*Dobnik et al., 2016*; *Klymus et al., 2019*). LoQ and LoD are generally determined using dilution series of synthetic DNA, as it can be less variable than gDNA, allowing for interlaboratory comparisons of assay performance (*Langlois et al., 2020*; *Thomson-Laing et al., 2021*; *Xia et al., 2021*). In this study, we designed gBlocks™ (Integrated DNA Technologies, Coralville, IA, USA) for a 700-bp gene region covered by the assay, with specific primer and probe binding sites for the four target species. To differentiate synthetic DNA from tissue-derived DNA in the event of cross-contamination, each gBlock was modified by reversing a 12-bp fragment between the forward and reverse primers of the assay, without overlapping into the probe region. The optimized tetraplex ddPCR™ assay was evaluated for sensitivity on a ten-fold series of 10X dilutions from a starting gBlock concentration of 2 ng/µL, in triplicate reactions. Target DNA was detected in all replicates for all target species down to $2 \times 10^{-8}$ ng/µL. At the $2 \times 10^{-9}$ ng/µL dilution, target DNA was detected in one or two of the three replicates across all four species. No target DNA was detected in the $2 \times 10^{10}$ ng/µL dilutions.

To further refine the LoQ and LoD, ddPCR™ reactions were performed on a series of 2X dilutions consisting of $1 \times 10^{-8}$, $0.5 \times 10^{-8}$ and $0.25 \times 10^{-8}$, and on $2 \times 10^{-9}$ and $1 \times 10^{-9}$.

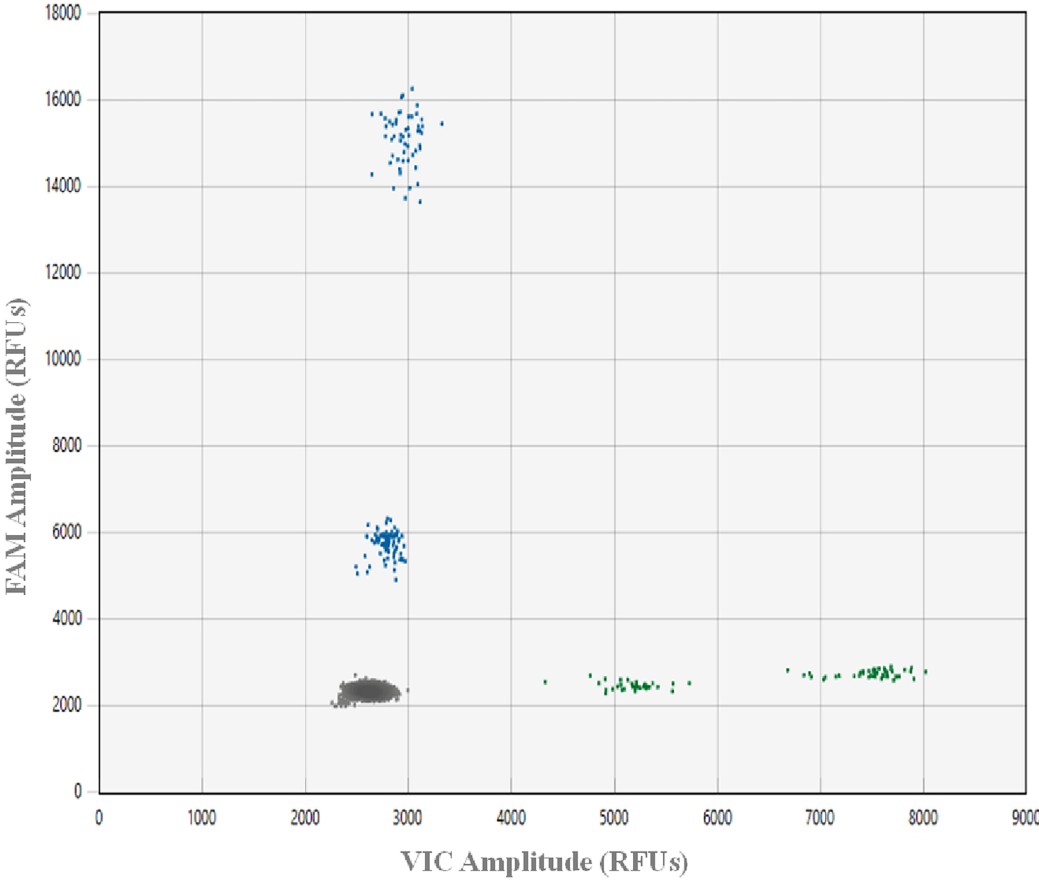

**Figure 1 Plot showing ddPCR$^{TM}$ droplet amplitude.** Values are presented in relative fluorescence units (RFUs), using 0.02 ng gDNA from greater amberjack *Seriola dumerili* (GAJ), lesser amberjack *S. fasciata* (LA), almaco jack *S. rivoliana* (AJ) and banded rudderfish *S. zonata* (BR) with the optimized assay conditions. Each droplet was classified as negative (grey in bottom left corner) or positive (blue on FAM Y-axis and green on VIC X-axis) using a Bio-Rad® QX200$^{TM}$ droplet reader, QX manager 1.2 standard edition software and the absolute quatification analysis setting.

Because replicates with no detections add significant variability, most ddPCR$^{TM}$ assays establish their LoQ at the lowest copy number in the dynamic range where ≥90–100% of replicates are detected, or the coefficient of variation (CV) of the measured copy number ≤25–35% (*Dobnik et al., 2016*; *Mauvisseau et al., 2019*; *Thomson-Laing et al., 2021*). In this study, target DNA of all four species was efficiently (Fig. 2) detected in all three replicates down to $0.5 \times 10^{-8}$ ng/μL, however at this lower end the CV was ≥55% across the four target species. For this reason, we set the LoQ at $1 \times 10^{-8}$ ng/μL corresponding to 0.35 copies/μL with a CV of 25% for *S. dumerili*, 0.64 copies/μL with a CV of 22% for *S. fasciata*, 0.47 copies/μL with a CV of 42% for *S. zonata*, and 0.49 copies/μL with a CV of 30% for *S. rivoliana*. Estimated eDNA concentrations below the LoQ should only be evaluated qualitatively, as detections or non-detections (*Klymus et al., 2019*).

QPCR guidelines (*Bustin et al., 2009*; *Klymus et al., 2019*) have been useful in guiding ddPCR$^{TM}$ LoQ, and the Minimum Information for the Publication of Digital PCR

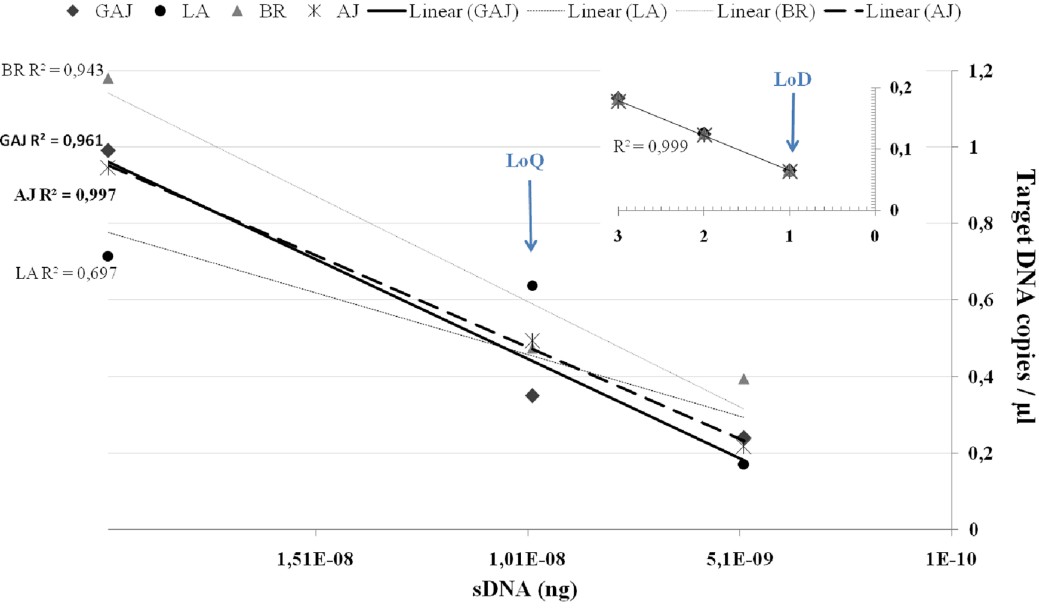

**Figure 2 Average target DNA concentrations (copies/µL) and ddPCR efficiency of a dilution series from a concentration of $2 \times 10^{-8}$ ng of sDNA (gBlocks) to one droplet.** Data is presented for greater amberjack *Seriola dumerili* (GAJ), lesser amberjack *S. fasciata* (LA), almaco jack *S. rivoliana* (AJ) and banded rudderfish *S. zonata* (BR). Smaller graph on top right corner shows ddPCR efficiency on one, two and three positive droplets obtained below the LoQ at $0.25 \times 10^{-8}$, $2 \times 10^{-9}$ and $1 \times 10^{-9}$ ng of sDNA. Assay LoQ and LoD are indicated by blue arrows.

Experiments (dMIQE, *Huggett et al., 2013*) has been recently updated (*Huggett & The dMIQE Group, 2020*). However, clear guidelines are needed for ddPCR™ LoD determination to maximize the benefit of using a highly sensitive platform. The power to detect as little as a single copy of target DNA in one amplified droplet is a major strength of ddPCR™ and the driver behind the increased application of (including transfer of qPCR assays to) ddPCR™ in eDNA research (*Hunter et al., 2018*; *Dimond et al., 2022*; *King et al., 2022*). In this study, to establish much needed reasoning and confidence in the 'one-droplet-power' of ddPCR™, we proposed a new approach to LoD. We also include efficiency curves which, although not needed for quantification purposes in ddPCR™, are still a useful performance indicator. In ddPCR™, below the LoQ and when there are far fewer target DNA copies than droplets, a single droplet partition will contain no more than one target DNA copy at the start of the reaction (BioRad Droplet Digital PCR Applications Guide Bulletin 6407). Because each droplet represents its own PCR reaction, ddPCR™ amplification efficiency, accuracy, repeatability, and overall reliability can be (re) determined at the droplet level (Fig. 2). Below the LoQ, in the $0.25 \times 10^{-8}$, $2 \times 10^{-9}$, and $1 \times 10^{-9}$ dilutions, one, two, and three ddPCR™ droplets were detected in at least one replicate across all four species. For these one, two, and three droplets detections, ddPCR™ efficiency was high $R^2 = 0.999$ and the 'one droplet/target DNA copy' LoD corresponded to $0.065 \pm$ SE 0.003 target DNA copies/µL for *S. dumerili*, $0.063 \pm$ SE 0.002 copies/µL for *S. fasciata*, $0.067 \pm$ SE 0.003 copies/µL for *S. zonata* and $0.064 \pm$ SE 0.001 copies/µL for *S. rivoliana* (Fig. 2).

## Field surveys

### Sampling gears and vessel preparation

To validate the developed ddPCR$^{TM}$ assay *in situ* and groundtruth eDNA field data, seawater sampling was conducted during a fisheries survey gear calibration cruise from August 29$^{th}$ to September 2$^{nd}$, 2022. Twenty-four hours prior to fieldwork, a deep clean was conducted of the research vessel to prevent contamination from previous cruises. Exterior working areas were sprayed with 10% bleach for ≥20 min, followed by a thorough freshwater rinse. Inside areas were cleaned with 10% bleach wipes. Near-concurrent eDNA, acoustic, and camera imagery data were collected from baited chevron trap cameras, stereo baited remote underwater video (S-BRUV), a remotely operated vehicle (ROV), and virtual positioning system (VPS) arrays. Camera and eDNA gears were allocated approximately two hours each for sampling and, for logistic reasons, distributed on two vessels (Fig. 3). We have acknowledgement from NOAA to conduct scientific research on approved vessels in accordance with the definitions and guidance at 50 CFR Part 600.10 and 600.745(a) and permit F/SER24:RM. As such, the activities are not subject to fishing regulations at 50 CFR Part 622 or otherwise developed in accordance with the Magnuson-Stevens Fishery Conservation and Management Act. The cameras and passive acoustic devices employed in this study were used only underwater with respect to privacy, and no people were underwater within the range of our camera or acoustic gears.

### Site selection, fish tagging, and VPS arrays

All gears were deployed at two artificial reef 'super pyramids' (∼8 m tall with a 4.5 m base) in ∼30 m depths off the coast of Dauphin Island, Alabama (AL), U.S. (site 1 at 29° 52.785 N −87° 58.328 W and site 2 at 29° 47.725 N −87° 57.245 W). These two sites were chosen due to the relatively large number of *S. dumerili* present, and the success of acoustic tagging efforts. From August 21$^{st}$ to 26$^{th}$, 2022, *S. dumerili* (500–1,175 mm TL, median size of 1,020 mm) were captured using standard hook and line gear. Fish in good condition were affixed with acoustic transmitters ($n$ = 41; Innovasea V16P, V9P, or V9, random delay from 60–120 s) and conventional tags (Floy Tag BFIM-96, double-head dart). On August 26$^{th}$, 2022, calibration staff installed a VPS array at both artificial reef sites to groundtruth eDNA and camera gear data. The VPS arrays consisted of a ring of inner receivers (Innovasea VR2AR, $n$ = 4) placed approximately 125 m from the reef, while an outer array of receivers ($n$ = 4) was placed ∼250 m from the inner receivers, resulting in ∼1 km$^2$ of VPS coverage between both sites. The two sites were visited on alternate days by both vessels, and gear deployment order and sampling vessel were randomly selected all five days of the calibration cruise (Fig. 3). All work was conducted under tagging protocol number 2129232 entitled 'Monitoring movements of fishes around coastal and offshore Alabama: Fish biotelemetry and biologging', as approved by the University of South Alabama Institutional Animal Care and Use Committee (IACUC).

### Camera gear operations

Each camera sampling gear (ROV, S-BRUV, chevron trap mounted cameras) was deployed following standard survey-specific sampling protocols. ROV sampling followed

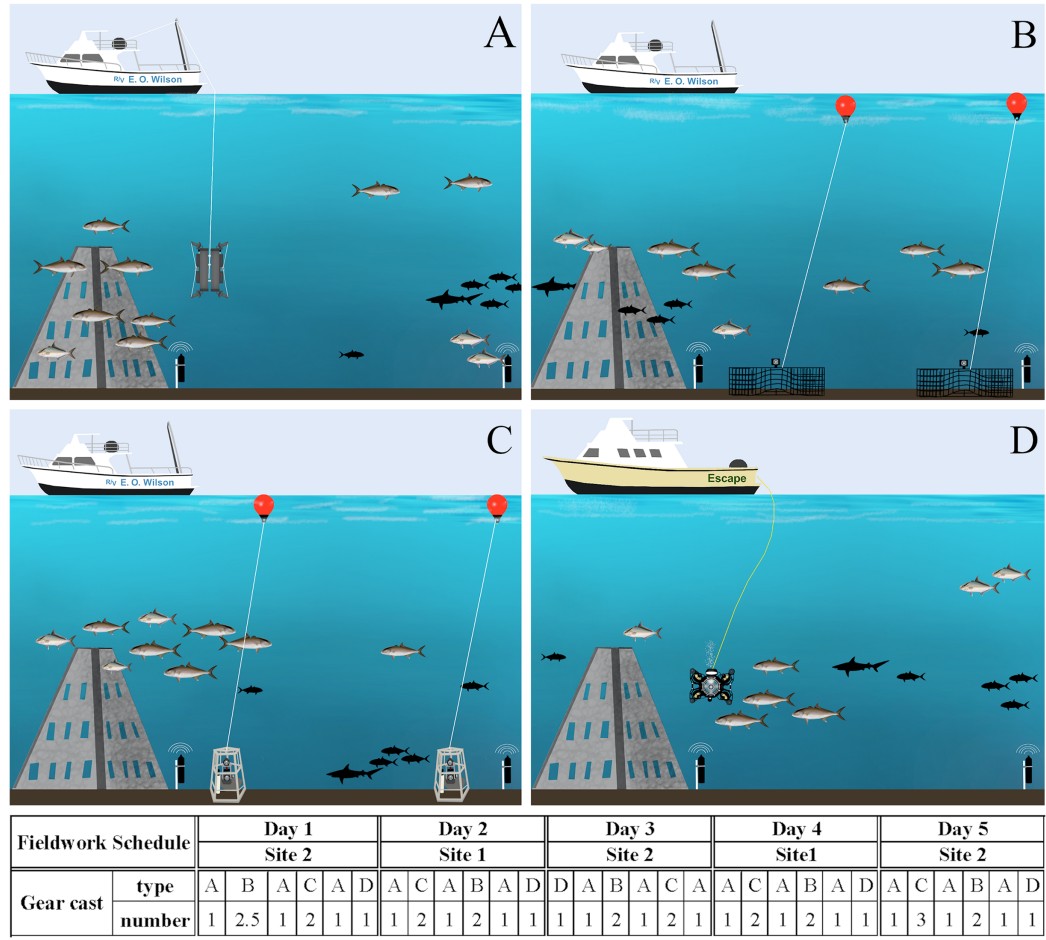

| Fieldwork Schedule | | Day 1 | | | | | | Day 2 | | | | | | Day 3 | | | | | | Day 4 | | | | | | Day 5 | | | | | |
|---|---|---|---|---|---|---|---|---|---|---|---|---|---|---|---|---|---|---|---|---|---|---|---|---|---|---|---|---|---|---|---|
| | | Site 2 | | | | | | Site 1 | | | | | | Site 2 | | | | | | Site1 | | | | | | Site 2 | | | | | |
| Gear cast | type | A | B | A | C | A | D | A | C | A | B | A | D | D | A | B | A | C | A | A | C | A | B | A | D | A | C | A | B | A | D |
| | number | 1 | 2.5 | 1 | 2 | 1 | 1 | 1 | 2 | 1 | 2 | 1 | 1 | 1 | 1 | 2 | 1 | 2 | 1 | 1 | 2 | 1 | 2 | 1 | 1 | 1 | 3 | 1 | 2 | 1 | 1 |

**Figure 3 Fieldwork setup scheme.** Gear deployment for eDNA (A), chevron traps (B) and S-BRUVs (C) on R/V E.O. Wilson (vessel 1), and ROV (D) on the Escape (vessel 2). The different gear types were deployed during 5 days on two sites following the order and number of casts in the schedule below the figures. Illustration credit: Bryan Huerta-Beltrán.

*Powers et al. (2018)*. Briefly, video footage of the reef sites was recorded using high-definition video on a four-thruster ROV equipped with sonar and 360-degree viewing capabilities. At each site, the ROV was positioned ~5 m from the structure, 2 min of video were recorded, and the process was repeated on the opposite side of the structure for an additional 2 min. The ROV was then positioned approximately 1 m above the reef to record a 360-degree view of the area, before being retrieved manually by attachment to a tagline or hand hauled on board.

S-BRUV sampling consisted of two pods deployed in succession using the vessel cat head. Each pod consisted of two independent stereo-video recorders (each with two digital video cameras recording 10 images per second) mounted opposite each other inside a 0.6 × 0.6 × 0.6 m aluminum frame. Additionally, GoPro cameras were mounted orthogonal to the stereo-video recorders to provide habitat view in all directions. Pods were randomly selected to be baited (two mackerel *Scomber* spp. halves and two squid) or unbaited, and were deployed for 35 min (*Keenan et al., 2018*). Chevron trap camera deployment followed

*Bacheler et al. (2020)*. Two chevron traps (dimensions 4′ × 5′ × 2′) were baited with 24 Gulf menhaden (*Brevoortia patronus*) and soaked for approximately 40 min. Traps were deployed ≥200 m from one another to ensure independence between trap samples and equipped with two GoPro cameras, over the mouth and over the nose of the trap. Only cameras attached over the mouth of each trap were used to count *Seriola* species and videos were excluded if the traps bounced, videos were too dark or out of focus to identify fish species, the camera view was obstructed, or the video files were corrupt.

## eDNA sample collection and filtration

To account for the potential impact of the baited survey gears on eDNA data, seawater sampling was conducted each day at three timepoints: before, between, and after S-BRUV and chevron trap deployments (Fig. 3). Seawater sampling for eDNA analysis occurred as close as possible to the other gears deployment/retrieval locations, and all time and GPS coordinates were recorded from the vessel GPS unit. During each sampling timepoint, two duplicate Niskin samplers attached to a metal frame were deployed using the vessel winch line and A-frame, followed by a Sea-Bird SBE25 CTD to collect environmental parameters (salinity, temperature, oxygen) along the depth profile. A total of fifteen seawater samples were collected in duplicate (30 × 10 L) over the 5 days of the cruise.

The Niskin samplers were cleaned with 10% bleach and rinsed with deionized water before, between, and after deployments. To test for contamination during sample collection, 10 L of autoclaved deionized water were processed through each Niskin sampler and filtered at the end of each sampling day. During the first four days of sampling, seawater samples were emptied into a sterile 10 L jerry can and kept at −20 °C until filtration, which occurred within 18 h of collection in a designated water filtration laboratory, where fish tissue had never been present. Water samples were filtered using a bench top GAST® DOA-P704-AA vacuum pump (Gast, Benton Harbor, MI, USA) and a self-preserving filter housing (Smith-Root, Vancouver, WA, USA; *Thomas et al., 2019*) fitted with Whatman® (Maidstone, United Kingdom) 47 mm 1.2 micron (µm) nitrocellulose filters, which were replaced when clogged (1–2 filters were used for each 10 L water sample). On the last sampling day, seawater samples were filtered at-sea directly from each Niskin using a Grover-Go portable vacuum pump (Grover Scientific, Rosslea, QLD, Australia) and the Smith-Root self-preserving filter housings. All filter housings were stored short-term (≤6 months, *Thomas et al., 2019*) at room temperature per manufacturers' protocols until eDNA extraction.

## eDNA extraction, ddPCR™ screening and data analysis

Total eDNA was extracted from a half portion of each filter using the QIAGEN® DNeasy® Blood & Tissue Kit (Hilden, Germany) following the *Goldberg et al. (2011)* protocol incorporating QIAshredder™ spin columns and a final elution step of 50 µL. During eDNA extraction, designated sterile forceps and blades were used to handle filters for each sample to prevent cross-contamination. Extraction negative controls were processed alongside the field samples but contained only reagents. Field negative controls were extracted, screened, and deemed negative for contamination before eDNA extraction

of samples occurred. The optimized tetraplex ddPCR™ assay developed in this study was used to screen all eDNA samples and negative controls (field, extraction, PCR) for the presence of the four target species using 4.5 µL of template per reaction, in replicates of three. All remaining half filters and eDNA extracts were stored at −80 °C.

All ddPCR™ reactions were run in the BioRad QX200 AutoDG™ Droplet Digital™ PCR System and data were analyzed using Absolute Quantification (ABS) and the QX Manager 1.2 Standard Edition software, following guidelines on the BioRad Droplet Digital PCR Applications Guide Bulletin 6407. Field samples and controls were analyzed using a three-criterion approach to reduce the likelihood of false positives (*Schweiss et al., 2020*; *Lehman et al., 2020*). Samples were defined as positive detections if, in at least one ddPCR™ replicate per sample, droplet(s) were above the threshold established for NTCs, within the normal range of positive droplets for the assay, and the concentration of target DNA was at or above the LoD. For controls to be considered free from contamination, none of the criteria above could be met.

## Camera gear and VPS array data analysis

MaxN for each *Seriola* species and genus level MaxN for *Seriola* were determined for each ROV, S-BRUV and chevron trap deployment. For all camera types, MaxN for each species was defined as the maximum number of individuals of each *Seriola* species in any frame within the video, and genus level MaxN was defined as maximum number of *Seriola* in any frame in the video. Data from the VPS array were downloaded and processed using Innovasea Fathom software to calculate virtual positions (latitude and longitude) of acoustically tagged fish within the arrays across the 5-day sampling period. The abundance of *S. dumerili* at the two sites was then estimated using a mixed logit-normal mark-resight model (Program MARK). For model inputs, the number of acoustically tagged fish in the core region (within approximately 125 m from reef) at each site was determined from the VPS position data, and we used MaxN counts of confirmed tagged and confirmed untagged individuals from the ROV videos for the resighting events. The numbers of tagged fish seen on the other camera gears was too low and inconsistent to use for the analysis.

## Statistical analysis

We conducted Fisher's exact tests to evaluate whether or not detection probabilities were statistically distinguishable between all the gears used. To quantify the relationship between quantitative eDNA estimates of *S. dumerili* with other acoustic and camera gears abundance measures, we calculated the Spearman correlation using a Bonferroni correction to adjust for the multiple comparisons. Spearman correlation is a nonparametric measure of the relationship between two measures and is calculated as the Pearson correlation of ranks of the measures rather than their values, which can be useful when data are skewed or have outliers. It was chosen instead of the Pearson correlation to describe these relationships among the abundance measures due to robustness of its statistical properties to lack of normality, an assumption required for statistical tests of Pearson correlation.

## RESULTS

### Assay development

Fish mtDNA COI sequences are widely available in GenBank, representing a good starting point for assay development. In this study however, we found design options to be more limited for this gene and primers less specific during both *in silico* and *in vitro* testing. Assays designed for the mtDNA ND5-ND6 region were primarily based on the 39 sequences generated by *Renshaw & Gold (2009)* for the four target *Seriola* species in the U.S. Atlantic Ocean and Gulf of Mexico (GenBank accession numbers GU014709–GU014747). Although few mtDNA ND5-ND6 sequences are available on GenBank, whole mtDNA sequences are common and primers did not match any non-target marine species, or common organisms (*i.e.*, human, bacteria) that can represent a source of exogenous DNA contamination. From the two ND5-ND6 primer sets we designed, one cross-amplified DNA from all exclusion species at <40 PCR cycles, even at higher annealing temperatures ($T_a$ 60 °C), and was therefore eliminated. The selected primer set (Table 2) successfully amplified the target locus in *Seriola* spp. but did not cross-amplify DNA from exclusion species. The addition of the four species-specific probes provided further specificity of the assay, with each amplifying DNA from a single target *Seriola* species. From all exclusion species (Table 1) tested, only *S. lalandi* produced non-specific amplification droplets *via* the *S. dumerili* probe.

In this work, designing primers and probes on multiple gene regions provided a better chance at developing an efficient and effective assay, while initially testing primers using qPCR lowered costs. Real-time amplification curves, Ct values, and melting curves can provide an initial straightforward visual assessment of amplification efficiency, an indication of cycles above which unwanted non-specific amplification occurs, as well as of the presence of potential primer-dimers. The highest amplitude of an artifact droplet across the 48 NTC ddPCR™ reactions was 500 RFUs above the average RFUs of the negative droplets cluster across the FAM and VIC channels. Therefore, to be conservative, 1,000 RFUs above the average RFUs of the negative droplets clusters was chosen as the NTC threshold for this assay (Fig. 1). The absence of dimers between oligos (primers and probes) in this assay facilitated adequate fluorescence amplitude between negative and positive droplet clusters for all target species. In addition, despite the ability to multiplex on the QX200 using the Bio-Rad® ddPCR™ supermix for probes, the use of the ddPCR™ Multiplex Supermix for probes (a concentrated version developed for the QX600) allowed for a 1,000–2,000 RFUs increase in probe amplitude.

In order to design the best possible assay, we tested two of the probe oligos using both FAM and HEX fluorophores. When compared to HEX, the FAM fluorophore increased probe RFUs by up to 12,000, which is in line with this being primarily advised for ddPCR™ experiments (BioRad Droplet Digital™ PCR Applications Guide Bulletin 6407). On the HEX/VIC channel, we found IDT SUN (VIC equivalent) fluorophores to provide slightly better amplitude differences between droplet clusters than HEX. However, detections on the HEX/VIC channel were never above 6,000 RFUs, limiting the fluorescence amplitude between the two target species on this channel. This same issue has

been reported in similar work, where droplets on the edge of the range for a target species could be incorrectly assigned to another species, resulting in false positive detections (*Dobnik et al., 2016*; *von Ammon et al., 2022*). To minimize this issue, we assigned distinct fluorophores for the detection of the two most common species *S. dumerili* and *S. rivoliana*. On the FAM channel, the optimal range of positive droplets for *S. dumerili* and *S. fasciata* was 2,500 to 4,500 RFUs and 12,000 to 14,000 RFUs above the average RFUs of the negative cluster, respectively. On the VIC channel, the optimal range of positive droplets for *S. zonata* and *S. rivoliana* was 1,500 to 3,500 RFUs and 4,000 to 6,000 RFUs above the average RFUs of the negative cluster, respectively (Fig. 1). To guard against false positive detections of *S. rivoliana*, when positive droplets generated on the VIC channel fell towards the lower end of its positive droplet RFU range, the ddPCR was repeated using only the *S. zonata* probe to rule out the presence of this species.

## eDNA field sampling data

Niskin samplers deployments took an average of 3 min, and eDNA 'gear' operations including the CTD cast never exceeded 15 min of the two hours allocated. Gear deployment distance from the reef varied from 4 to 92 m (average 31.5 ± SD 23.8 m), with the vessel drifting 17 to 84 m between deployment and retrieval (average 38.3 ± SD 23.16 m). Environmental parameters at the depth of sampling throughout all five sampling days presented averages of 35.8 ± SD 0.09 salinity (psu), 26.9 ± SD 0.25 temperature (°C), 4.2 ± SD 0.41 dissolved oxygen (mg/L) and 7.8 ± SD 0.03 pH (File S1).

Analysis of negative controls (field, extraction, PCR) found no evidence of target DNA across PCR replicates. However, we failed to include a filtration control in the laboratory, which means that if target DNA had been detected in the field negative controls, we would not have been able to determine if contamination had occurred during seawater sampling or filtering. Both processes were time consuming and prone to contamination during the first four days of sampling. On the fifth day of sampling, using a portable Grover pump for filtering the seawater samples on board directly from the Niskin samplers was faster, reduced the number of areas where samples were handled (vessel, storage freezer, laboratory) and equipment needed, greatly simplifying the process and reducing the potential for contamination.

From the 15 eDNA samples collected, 14 were positive for *S. dumerili* and 12 were positive for *S. rivoliana*. No positive detections were obtained for *S. fasciata* and *S. zonata*. eDNA detections before any baited fisheries gear entered the water were above the LoQ for one out of the five sampling timepoints for both *S. rivoliana* and *S. dumerili*, respectively (Fig. 4). eDNA collected between and after deployment of baited fisheries gears were above the LoQ for *S. dumerili* on eight of the 10 sampling timepoints. *Seriola rivoliana* eDNA was not detected in samples collected between baited gears on site 1 and while it was detected above the LoD each day at site 2, this detection was always below LoQ. Samples collected after all baited gears were deployed were above LoD for *S. rivoliana* at both sites and all sampling days, but were only above LoQ on the second day (Fig. 4).

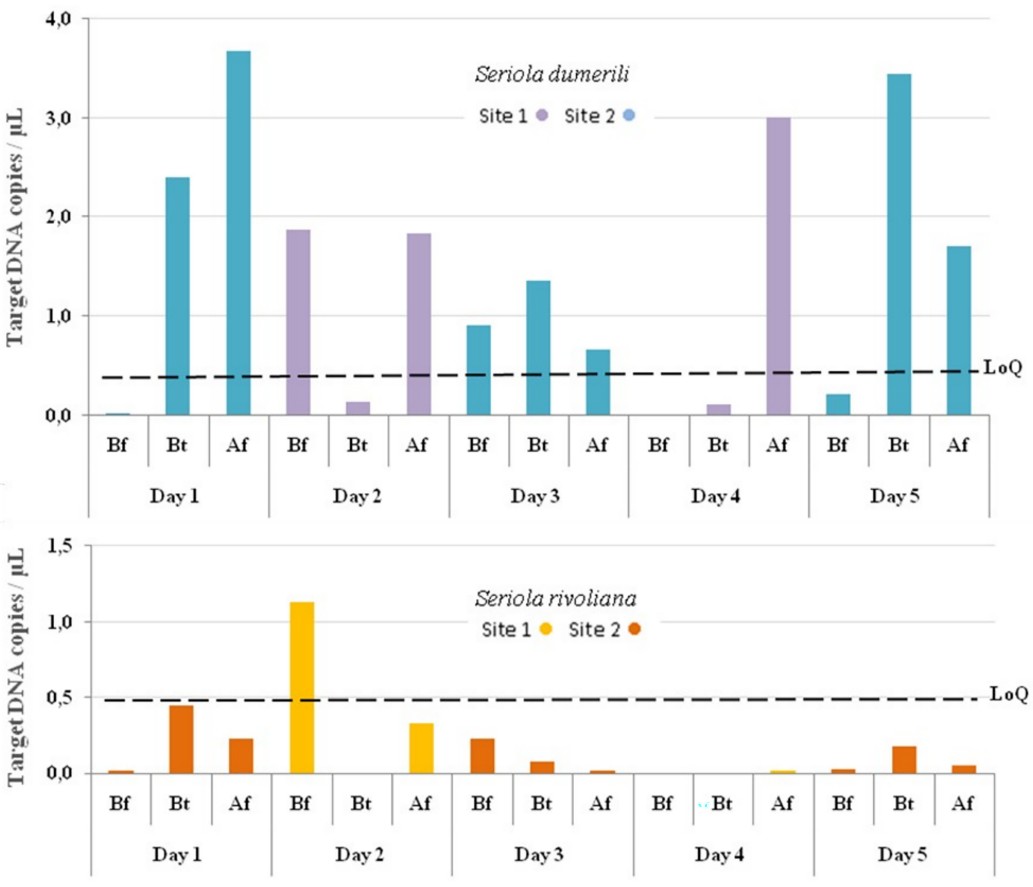

**Figure 4 Graphs showing *Seriola dumerili* and *S. rivoliana* ddPCR^TM detections from eDNA samples.** Seawater eDNA samples were collected during the 5 days of sampling at two sites (1 and 2) before (Bf), between (Bt) and after (Af) the deployment of chevron trap and S-BRUV gears (for details on fieldwork setup see Fig. 3). DdPCR detections are expressed as target DNA copies/μL of PCR reaction for *Seriola dumerili* and *S. rivoliana*, and the limit of quantification (LoQ) indicated by a dashed black line for both species (see Data S2 for data details).               

## Camera gear and VPS array data

All camera gears made full use of the 2 h allocated for sampling. All camera gears detected *S. dumeril* and *S. rivoliana* (when unsure these were assigned to *Seriola* genus level) at both sites and all 5 days of sampling (File S2), but did not detect *S. fasciata* or *S. zonata*. The highest MaxN counts observed during the study were 24 *S. dumerili* and four *S. rivoliana*. The VPS receivers confirmed that there were at least six *S. dumerili* on the sites throughout the 5 days. For site 2, eight individuals were present in the core region (within ~125 m of the target reef) during all 5 days. For site 1, six individuals were observed in the core region on the first day of sampling, and seven were observed in this region from day two to five of sampling.

## Statistical analysis

All the gears and eDNA timepoint measures were effective at detection of *S. dumerili* (File S2). All gears except SBRUV, and all eDNA timepoints except one (before) showed an

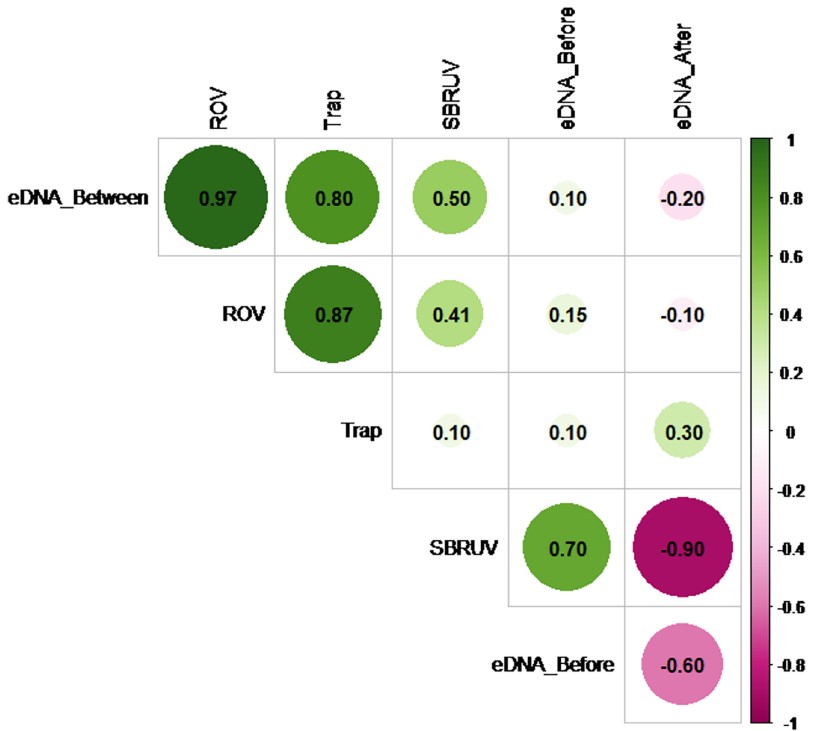

**Figure 5 Matrix correlation plot.** Showing Spearman correlation coefficient (r) among pairs of eDNA data collected at all three timepoints (before, between and after camera gear deployments) and camera gear data across all five sampling days. The magnitude of the correlations is depicted by color and shade (pink for negative and green for positive), and by size for magnitude of absolute value.

abundance measure greater than 0 on all five sampling occasions. The two exceptions detected the species on four of the five occasions. This difference (between 80% and 100% detectability on five sampling occasions) is not statistically distinguishable, as a Fisher's exact test fails to reject the hypothesis of equal detection probabilities ($p$-value = 1).

However, the similarity of the abundance measures themselves does vary. Figure 5 shows the Spearman correlation matrix calculated from the data of File S2 for the six measures of abundance available on the $n = 5$ sampling days. These include the three *S. dumerili* eDNA timepoints (before, between and after), ROV, Chevron trap and SBRUV measures of abundance.

The Spearman correlation matrix shows that the eDNA between timepoint, ROV, and Chevron Trap abundance measures are closely aligned with each other, while the other two eDNA timepoints show little, or even negative, relationships with the three abundance measures. Hypothesis tests to detect presence of positive correlations in any of the 15 pairs of measures were conducted, using a Bonferroni correction to adjust for the multiple comparisons. The highest correlation among any pair of measures (r = 0.97) is between the eDNA between timepoint and ROV measures. The hypothesis tests showed that this was the only statistic large enough to confirm a positive correlation at significance level $\alpha = 0.05$ (since its $p$-value = 0.0024 < 0.05/15 = 0.0033), likely as a result of the lack of power from the small sample size.

## DISCUSSION

In this study, we developed and validated a novel tetraplex ddPCR™ assay that is able to detect and distinguish between *S. dumerili*, *S. fasciata*, *S. rivoliana* and *S. zonata*. To our knowledge, this is the first ddPCR™ assay able to detect four aquatic species simultaneously in a single sample, and only the second published application of ddPCR™ in marine fisheries research (the first being *Maes et al., 2024*). This novel assay was highly specific *in silico* against sequences of other organisms in GenBank, *in vitro* against an extensive list of co-occurring and related exclusion species, and *in situ*, supported by datasets from multiple fisheries survey gears. There was no evidence of contamination during laboratory and field processing suggesting that detections are valid. Throughout this study, we have demonstrated that with strict cleaning and handling protocols and thorough testing of negative controls, it is viable to conduct robust eDNA research as part of a fisheries survey cruise. The validation of the species-specific assay for *S. dumerili*, *S. rivoliana*, *S. zonata* and *S. fasciata* in the field allowed for the development of a robust and efficient methodology for field use, notably the ability to quickly filter and preserve samples, thus limiting potential contamination that can result from multiple handling during these processes.

We do note that because the assay was developed using mtDNA loci alone, it will not be able to identify potential hybrids. However, at the time of this study, no nuclear sequences of *Seriola* species were available on Genbank and, although hybrids are known to occur in this genus (*Takahashi et al., 2021*), there is not yet evidence of hybridization among the target species of this study in U.S. waters. The assay specificity was tested on regionally co-occurring species, with the aim of making it suitable for eDNA research in the U.S. Atlantic Ocean and Gulf of Mexico. Further testing of co-occurring and closely related exclusion species must be conducted prior to using the assay on eDNA samples from other regions. The fact that the assay showed non-specific detection for *S. lalandi* is not relevant for the current work as this species does not occur in the U.S. Atlantic Ocean and Gulf of Mexico (*Swart et al., 2015*) but could be an issue in areas where it occurs. Because the primers developed amplify DNA across all *Seriola* species, the assay could be adjusted *via* probe redevelopment and optimization to detect species within this genus in other areas of the world. Sensitivity (LoQ and LoD) must be reported for any new markers and/or areas for the sake of assay transparency and reliability, and to allow future users to make informed decisions on whether to adopt existing markers or design new ones (*Xia et al., 2021*).

### Comparison of eDNA detection to gears detection data

Both eDNA and camera gears detected the presence of *S. dumerili* and *S. rivoliana*, but not *S. fasciata* and *S. zonata*. This result was expected, as *S. dumerili* and *S. rivoliana* are the most common species in the surveyed region of AL where *S. fasciata* (mostly present in the western Gulf of Mexico) and *S. zonata* (mostly present along the Atlantic) are rarely seen. *Seriola dumerili* eDNA detection data aligned with detections from acoustic and camera gears. *Seriola dumerili* were detected on all sampling days by the acoustic array, ROV and chevron trap cameras. *Seriola dumerili* were also detected on S-BRUV videos all days

except day four, but this could be explained by camera malfunctions that resulted in a lower sample size on this day. Similarly, *S. dumerili* were detected by eDNA during all days, and all sampling timepoints except one on day four that could be explained by the timing and distance of deployment. Specifically, this sample was collected before any baited gear was deployed and at the furthest distance (72 m) recorded from the reef. Nevertheless, similar detection patterns were observed across camera and acoustic gears on day four, with some of the lowest detections observed for *S. dumerili* compared to all other sampling days (File S2). After an overnight storm, we experienced increased currents on days three and four which resulted in a higher gear deployment distance from the reef on day four. This could explain the overall lower detections for *S. dumerili* as sampling distance could influence fish detection across all gears.

eDNA detection data for *S. rivoliana* also aligned well with overall detections from other gears. *Seriola rivoliana* were detected by ROV, chevron trap and S-BRUV cameras on all sampling days. *Seriola rivoliana* eDNA was also detected on all sampling days and at 12/15 sampling timepoints. Six of these 12 positive detections include 'one droplet' detections, demonstrating the importance of LoD and validating assay sensitivity in the field. *Seriola rivoliana* were detected after all baited gear deployments, but not between deployments on day two and four or before gear deployment on day four. Baited gear deployment is likely to have an influence on eDNA detection, given its ability to attract *Seriola* species.

## Comparison of eDNA concentration to gears abundance data

In this study, eDNA relative concentration data was higher for *S. dumerili* than *S. rivoliana* at both sites for all sampling days, in line with relative abundance patterns obtained from camera gears. Both *S. dumerili* and *S. rivoliana* eDNA relative concentrations were higher at site 2 compared to site 1. For *S. dumerili*, higher abundance was also observed on acoustic and camera gears at site 2 compared to site 1. For *S. rivoliana*, higher abundance at site 2 compared to site 1 was not observed across all camera gears, possibly due to the high variance inherent to fisheries gears (File S2).

The LoQ establishes the level above which eDNA concentrations can be reliably used for quantification comparisons to other gears and potential abundance estimates (*Dobnik et al., 2016*; *Klymus et al., 2019*). *Seriola dumerili* eDNA concentrations were above the LoQ at 10/15 sampling timepoints, across all sampling days. These included all timepoints after multiple baited camera gears were deployed, three out of five timepoints between gear deployments and two out of five timepoints before any gear was deployed, on days two and three (Fig. 4). However, on the third day of sampling, sampling vessel 2 visited the reef ahead of sampling vessel 1 (Fig. 3). Also, higher *S. dumerili* eDNA concentrations were observed on day one and day five, when baited camera gears were deployed more than two times each (Figs. 3 and 4). These results further suggest the deployment of bait and/or gears, and its ability to attract *Seriola* species, have an influence on eDNA concentrations.

*Seriola rivoliana* eDNA relative concentrations were above the LoQ at only one sampling timepoint, before gear deployment on day two. Higher eDNA concentration data before gear deployment on day two for both *S. rivoliana* and *S. dumerili* can result from the species proximity to the Niskin sampler. This was also noted on day four after all
deployments and where, despite sampling at the furthest distance to the reef (92 m), one of the highest *S. dumerili* eDNA relative concentration was obtained (Fig. 4).

For *S. dumerili*, eDNA concentrations above the LoQ allowed for a better understanding of the relationship to baited gear deployment and sampling distance from the reef. There were, however, contradictions in the data that highlight a challenge of eDNA, where *S. dumerili* proximity to the sampling device needs to be disentangled from higher fish abundance. Only by increasing sampling coverage and replication (by using *e.g.*, a rosette sampler, not available in this study) could target species eDNA be modeled over relevant gradients, disentangling proximity from abundance and reducing variance and uncertainty in eDNA data.

This study took place outside of the known *Seriola* sp. spawning season (March to June), making it unlikely that this biological factor could contribute to any contradictions observed. The size range of *S. dumerili* captured during the tagging cruise and the size range of *S. dumerili* and *S. rivoliana* observed on the cameras indicated young adult fishes to be mostly present. The fact that eDNA sampling took place immediately after the camera gears, limited the influence of eDNA decay with time and dispersal with currents, as much as possible. All other environmental parameters were fairly constant throughout the 5 days of sampling, and therefore unlikely to be responsible for any variation in eDNA data between sampling days.

## Incorporating eDNA into fisheries surveys

The non-invasive, relatively simple faster and overall cost-effective methodology represents some of the most compelling advantages of incorporating eDNA sampling in fisheries surveys. Assay specificity has also often shown eDNA to be a more reliable method for species detection than traditional survey methods (*Jerde, 2019*; *Ramírez-Amaro et al., 2022*; *Yao et al., 2022*). In the present study, eDNA field sampling was completed in significantly less time than camera gears. Among all gears, only VPS and eDNA provided 100% species-level detections (File S2). Acoustic arrays are mostly temporary, require continued investment to maintain, and are only able to detect tagged individuals, with the number of species and sample size of tagged fish being limited by the cost of purchasing and affixing acoustic transmitters. In contrast, ddPCR™ assay development represents a one-off initial investment for a given biogeographic area of sampling. In a ddPCR™ assay, individuals of at least four species can be detected without additional cost, and other species can be included with subsequent one-off partial investments.

The fact that baited gears seem to attract fish that are detected by eDNA suggests that it might be interesting exploring eDNA as a complementary method in surveys where baited gears are commonly used. All sampling methods have potential errors, and false absence (concluding the fish is absent when present) is known to occur in traditional fisheries sampling gears. eDNA can be a valuable complement to stationary camera gears (such as chevron trap and S-BRUV cameras) deployed in geographic areas and/or seasons with poor visibility conditions, where fish may be present but cannot be confidently identified to the species level.

Mobile camera gears such as ROVs are less constrained by deployment distance and visibility conditions, as they are controlled by an operator at all times and able to track habitat and fish. They are the gear of choice in fisheries surveys in Alabama, where they are known to provide reliable data for the type of reefs sampled in this study. The high correlation obtained between ROV and eDNA data collected between gear deployments adds to a growing number of studies demonstrating the potential of eDNA data as a cost-effective indicator of abundance for fisheries stock assessments (see review by *Rourke et al. (2022)*). eDNA data collected between gears are likely to represent a more realistic comparison to abundance data by any gear, while eDNA data collected after multiple baited gears deployment are likely to be inflated by recurrent attraction of fish.

Nevertheless, investment into follow-up studies with increased sample sizes and spatial and temporal replication would be necessary to allow for year-to-year comparisons that could validate the robustness of the correlation observed. A robust correlation fitted into a modeling approach, such as the one developed by *Chambert et al. (2018)* aimed at inferring animal density from eDNA and animal count data from a subset of sites, could be more confidently incorporated into existing fisheries management frameworks for Alabama. Time-series data from a carefully designed eDNA survey that estimates relative abundance could be used as an index of relative abundance for the *S. dumerili* stock assessment. eDNA holds potential for improving fisheries surveys as a reliable and cost-effective support 'gear'. Incorporating this gear with other common fisheries survey techniques would not require additional days at sea and could provide improved precision for estimated fish detection and abundance. Additionally, all archived eDNA samples can be used to investigate other species in future analyses, either by targeted assays or metabarcoding of fish assemblages (*Jerde, 2019*; *Ramírez-Amaro et al., 2022*; *Yao et al., 2022*).

## ACKNOWLEDGEMENTS

We would like to thank the time and effort kindly allocated by students and interns at the Mississippi State University's Marine Fisheries Ecology Program (Danielle McAree, Alena Anderson, Catlin Zimmer, and Alissa Pagel) and the USM Saillant lab (Tainá Rocha de Almeida) in sub-sampling fish tissue samples from their collections for this study. We also wish to thank to Mike Dance and Creed Branham at Louisiana State University for making samples of *S. fasciata* available, and Charlie Locke of the commercial fishing vessel *F/V Salvation* in NC for making samples of *S. zonata* available. We acknowledge the help of Belinda Swart and Rouvay Roodt-Wilding in pointing us to samples stored at Stellenbosch University in South Africa, obtained during their study (*Swart et al., 2015*) and provided by the following researchers: Mark Renshaw, Byron White, Kent Carpenter, Peter Wirtz, Haruhisa Fukada, H.J. Walker, Alastair Graham, Kate Hudson, T. Masumoto, Luca Castriota, Sven Kerwath, and Natalie Martinez-Takeshita. Thank you to Jeffrey Krause, Debbrota Mallick, Matt Boehm and Joshua Goff for valuable assistance during fieldwork planning and operation at DISL. Thank you to Kevin Dillon at USM Gulf Park Campus for lending the Niskin samplers for this fieldwork. Thank you also to the fantastic crew and colleagues on the *R/V E.O. Wilson*, particularly our vessel Captains (Jonathan Wittmann, Diana Marchant, and Michael Daudt) and volunteers from NEU (Savannah Swinea) and

USA Scyphers Lab (Sarah Gibbs). Last but not least, we thank USM graduate students Emma Humphreys and Annmarie Fearing for their valuable support during the initial deep cleaning of *DISL R/V* E.O. Wilson and the good vibes throughout.

### Funding

This research was funded by the U.S. Congress through the Mississippi-Alabama Sea Grant Consortium. The ddPCR$^{TM}$ work conducted in this study was supported by the Mississippi IDeA Network of Biomedical Research Excellence (INBRE), funded by an Institutional Development Award (IDeA) from the National Institute of General Medical Sciences of the National Institutes of Health under grant number P20GM103476. There was no additional external funding received for this study. The funders had no role in study design, data collection and analysis, decision to publish, or preparation of the manuscript.

### Grant Disclosures

The following grant information was disclosed by the authors:
U.S. Congress through the Mississippi-Alabama Sea Grant Consortium.
Mississippi IDeA Network of Biomedical Research Excellence (INBRE).
Institutional Development Award (IDeA) from the National Institute of General Medical Sciences of the National Institutes of Health: P20GM103476.

### Competing Interests

The authors declare that they have no competing interests.

### Author Contributions

- P. Joana Dias conceived and designed the experiments, performed the experiments, analyzed the data, prepared figures and/or tables, authored or reviewed drafts of the article, and approved the final draft.
- Ryan Lehman performed the experiments, authored or reviewed drafts of the article, and approved the final draft.
- Bryan L. Huerta-Beltrán performed the experiments, prepared figures and/or tables, authored or reviewed drafts of the article, and approved the final draft.
- Ana Wheeler performed the experiments, analyzed the data, prepared figures and/or tables, authored or reviewed drafts of the article, and approved the final draft.
- Crystal L. Hightower performed the experiments, analyzed the data, authored or reviewed drafts of the article, and approved the final draft.
- Jessica Heise performed the experiments, authored or reviewed drafts of the article, and approved the final draft.
- Theodore Switzer performed the experiments, authored or reviewed drafts of the article, and approved the final draft.
- Clint Rhode performed the experiments, authored or reviewed drafts of the article, and approved the final draft.

- J. Marcus Drymon performed the experiments, authored or reviewed drafts of the article, and approved the final draft.
- Lynne Stokes analyzed the data, prepared figures and/or tables, authored or reviewed drafts of the article, and approved the final draft.
- Mark A. Albins analyzed the data, prepared figures and/or tables, authored or reviewed drafts of the article, and approved the final draft.
- Sean Powers conceived and designed the experiments, authored or reviewed drafts of the article, and approved the final draft.
- Nicole Phillips conceived and designed the experiments, authored or reviewed drafts of the article, and approved the final draft.

### Animal Ethics

The following information was supplied relating to ethical approvals (*i.e.*, approving body and any reference numbers):

University of South Alabama Institutional Animal Care and Use Committee (IACUC) approved tagging protocol 2129232 on Monitoring movements of fishes around coastal and offshore Alabama: Fish biotelemetry and biologging.

### Field Study Permissions

The following information was supplied relating to field study approvals (*i.e.*, approving body and any reference numbers):

We have permission from NOAA to conduct scientific research on approved vessels in accordance with the definitions and guidance at 50 CFR Part 600.10 and 600.745(a). As such, the activities are not subject to fishing regulations at 50 CFR Part 622 or otherwise developed in accordance with the Magnuson-Stevens Fishery Conservation and Management Act. The cameras and passive acoustic devices employed in this study are used only underwater, we respect privacy and no people are found underwater within the range of our camera or acoustic gears.

### Data Availability

The raw data is available in the Supplemental Files.

### Supplemental Information

Supplemental information for this article can be found online at http://dx.doi.org/10.7717/peerj.18778#supplemental-information.

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
