# Peer review of "A novel ddPCR™ assay for eDNA detection and quantification of Greater Amberjack Seriola dumerilli and three congeners in US waters: challenges and application to fisheries independent surveys"

_PeerJ, doi:10.7717/peerj.18778_

## Round 0.1 · original submission · Major Revisions

I agree with the reviewers that although the manuscript highlights the use of ddPCR eDNA in species identification within the water column, and is especially beneficial for fish stock assessment, management and conservation, in addition to taxonomic identification, concerns regarding statistical analyses are being raised. I hope the authors could address each comment/suggestion and I look forward to reading the revised version of the manuscript.

Reviewer 1 ·

Basic reporting

The authors present the study ‘A novel ddPCRTM assay for eDNA detection and quantification of Greater Amberjack Seriola dumerilli and three congeners in US waters: challenges and application to fisheries independent surveys’. The manuscript give a significant advancement in the application of Droplet Digital PCR (ddPCR) for eDNA detection and quantification of Greater Amberjack and its congeners in the U.S. waters. It is a well-written study with clearly outlined objectives, rigorous methodology, and thorough results. However, some areas require further clarification and improvement to enhance the manuscript's scientific robustness and clarity.

Experimental design

The experiment is well-designed, the original primary research is fall within the aims and scope of the journal. Besides, The research questions were well defined, relevant and meaningful. The methods have been described accordingly.

Validity of the findings

the conclusion are well stated and connected to the original question investigated.

Annotated reviews are not available for download in order to protect the identity of reviewers who chose to remain anonymous.

Reviewer 2 ·

Basic reporting

The introduction contains more information (history) about the fishery than necessary for the study, but I found it very interesting and I hesitate to advocate for change. Please consider condensing L43-76. Similarly, the eDNA overview is unnecessary in the current scope, and lines 77-120 could be reduced. For example, most in the field have heard of ddPCR and its use in eDNA surveys for almost ten years. I do support acknowledging it has been underused in marine systems, and this work is a good example of application. The remainder of the manuscript adequately reports necessary information, but for some statistical interpretation of correlation that I discuss in general comments.

Experimental design

The experimental design has well described lab and assay procedures. I have no concerns about field collection methods and effort.

Validity of the findings

L:85 Relative abundance is still considered “potentially” correlated with eDNA concentrations from qPCR and ddPCR. Many variables, such as fish behavior, currents, and environmental conditions, make the hope for relative abundance tenuous in natural environments. For example, when do these fish spawn, and is it concurrent between species? Spawning releases pulses of high-quality DNA, much of which will not result in meaningful relative abundance estimates. See Takeuchi et al. 2019. Do these species hybridize? Mitochondrial DNA will reflect only material identification. Is this an issue in this system, and would this explain the variability in the eDNA results compared to taxonomic ID? See Takahashi et al. 2022 for an example from Japan. Ultimately environmental conditions can preserve or degrade eDNA at different rates and within year correlations to relative abundance from convention surveys hold, but year to year comparisons may not. This study does not provide a time series evaluation of the later and statements about the robustness of correlation should be worded carefully.

Additional comments

New approaches such as nuclear eDNA may hold more promise for population genetics evaluations of effective population size and fishery health. I encourage to the authors to provide a paragraph in the discussion about the findings of Adams et al. 2019, Andres et al. 2021, and Jo et al. 2022). qPCR/ddPCR have more limitations using the mitochondria than nuclear, but nuclear DNA is less abundant. Given recent advances, this point must be addressed.

Figures 2 and 3 are very hard to read. Please remove background color, increase dot size, and increase text size throughout all figures.

Figure 4 has an issue with dark colors not allowing for black text to be readable. Statistical tests of the difference of the correlation coefficient from zero should be reported for each correlation. Correlation coefficients without p-values or some measure of uncertainty, such as a confidence interval, are meaningless. Consider Bonferroni adjustment for study-wide significance. http://www.sthda.com/english/wiki/correlation-test-between-two-variables-in-r
And here
https://rpubs.com/JLLJ/SPC12B (reference within).


Adams, C. I., Knapp, M., Gemmell, N. J., Jeunen, G. J., Bunce, M., Lamare, M. D., & Taylor, H. R. (2019). Beyond biodiversity: Can environmental DNA (eDNA) cut it as a population genetics tool?. Genes, 10(3), 192.

Andres, K. J., Sethi, S. A., Lodge, D. M., & Andrés, J. (2021). Nuclear eDNA estimates population allele frequencies and abundance in experimental mesocosms and field samples. Molecular Ecology, 30(3), 685-697.

Jo, T. S., Tsuri, K., & Yamanaka, H. (2022). Can nuclear aquatic environmental DNA be a genetic marker for the accurate estimation of species abundance?. The Science of Nature, 109(4), 38.

Takahashi, H. (2022). Recent distributional shifts and hybridization in marine fishes of Japan. In Fish Diversity of Japan: Evolution, Zoogeography, and Conservation (pp. 311-325). Singapore: Springer Nature Singapore.

Takeuchi, A., Iijima, T., Kakuzen, W., Watanabe, S., Yamada, Y., Okamura, A., ... & Tsukamoto, K. (2019). Release of eDNA by different life history stages and during spawning activities of laboratory-reared Japanese eels for interpretation of oceanic survey data. Scientific reports, 9(1), 6074.

Reviewer 3 ·

Basic reporting

Comments in document attached

Experimental design

Comments in document attached

Validity of the findings

Comments in document attached

Additional comments

Comments in document attached

Annotated reviews are not available for download in order to protect the identity of reviewers who chose to remain anonymous.

---

## Round 0.2 · Major Revisions

Please kindly address the critical issue raised by reviewer 2, thank you.

Reviewer 1 ·

Basic reporting

the authors revised well and the manuscript can be accepted.

Experimental design

no comment

Validity of the findings

no comment

Reviewer 2 ·

Basic reporting

Addressed in revision. No comment.

Experimental design

Addressed in revision. No comment.

Validity of the findings

I must push back on the responses to reviewers' comments on the correlation analysis since this is a critical finding of the manuscript. I am good with the Spearman approach. The lack of power invalidates much of the inferences of this manuscript about the correlation of abundance estimates. Only one pair of measurements is significant (and not marked in the figure/matrix with an asterisk as stated in the response). Additionally, the experimental-wide error is incorrect without the Bonferroni adjustment (there are 15 tests being conducted in figure 5). The authors put this forward as "additional complexity." I posit this is a necessary complexity given the number of comparisons being conducted. This one positive is likely be due to Type 1 error. In its current form, I can not support including any of the correlation analysis. I am not convinced that eDNA between measures aligns well with the ROV.

Reviewer 3 ·

Basic reporting

no comment

Experimental design

no comment

Validity of the findings

no comment

Additional comments

no comment

---

## Round 0.3 · accepted · Accept

Thank you to all authors for following through the revision process and addressed all raised concerns meticulously.

Reviewer 2 ·

Basic reporting

no comment

Experimental design

no comment

Validity of the findings

no comment

Additional comments

no comment